# FedNAR: Federated Optimization with Normalized Annealing Regularization

**Junbo Li**[1], **Ang Li**[2], **Chong Tian**[1], **Qirong Ho**[1], **Eric P. Xing**[1,3,4], **Hongyi Wang**[3]

1. Mohamed bin Zayed University of Artificial Intelligence
2. University of Maryland 3. Carnegie Mellon University 4. Petuum, Inc.

## Abstract

Weight decay is a standard technique to improve generalization performance in modern deep neural network optimization, and is also widely adopted in federated learning (FL) to prevent overfitting in local clients. In this paper, we first explore the choices of weight decay and identify that weight decay value appreciably influences the convergence of existing FL algorithms. While preventing overfitting is crucial, weight decay can introduce a different optimization goal towards the global objective, which is further amplified in FL due to multiple local updates and heterogeneous data distribution. To address this challenge, we develop *Federated optimization with Normalized Annealing Regularization* (FedNAR), a simple yet effective and versatile algorithmic plug-in that can be seamlessly integrated into any existing FL algorithms. Essentially, we regulate the magnitude of each update by performing co-clipping of the gradient and weight decay. We provide a comprehensive theoretical analysis of FedNAR's convergence rate and conduct extensive experiments on both vision and language datasets with different backbone federated optimization algorithms. Our experimental results consistently demonstrate that incorporating FedNAR into existing FL algorithms leads to accelerated convergence and heightened model accuracy. Moreover, FedNAR exhibits resilience in the face of various hyperparameter configurations. Specifically, FedNAR has the ability to self-adjust the weight decay when the initial specification is not optimal, while the accuracy of traditional FL algorithms would markedly decline. Our codes are released at https://github.com/ljb121002/fednar.

## 1 Introduction

FL has emerged as a privacy-preserving learning paradigm that eliminates the need for clients' private data to be transmitted beyond their local devices [1, 2, 3]. FL presents challenges in addition to traditional distributed learning, including communication bottlenecks, heterogeneity in hardware resources and data distributions, privacy concerns, etc [4]. Numerous machine learning applications necessitate training in FL. For instance, hospitals may wish to cooperatively train a predictive healthcare model, but privacy regulations may mandate that each hospital's data remains local [5].

In traditional distributed optimization, local gradients are calculated for each client per iteration and subsequently sent to a global server for amalgamation. However, the communication constraints within Federated Learning (FL) entail multiple model updates prior to the aggregation phase at the global server, as exemplified by the widely-employed FedAvg framework [1]. Considering the skewed data distribution and multiple local updates for communication efficiency, it's pivotal to guard against overfitting models to local data. Algorithms like FedProx [4] and SCAFFOLD [6] were designed to address this issue. They build upon the FedAvg approach by integrating regularization terms during local training and employing weight decay in their local optimizers to combat overfitting on local clients. However, a frequently overlooked trade-off exists: local weight decay introduces an optimization objective that differs from the global one, an issue magnified in FL due to the numerous

37th Conference on Neural Information Processing Systems (NeurIPS 2023).

local updates from clients with varied data distributions. As demonstrated in Figure 1, a weight decay just slightly larger than optimal can result in an approximate 10% drop in accuracy. This poses a compelling research question:

*How can we strike a balance between averting overfitting on clients using weight decay, and minimizing update divergence in FL?*

In this study, we acknowledge the significant role weight decay coefficients play in FL algorithms and undertake an exhaustive examination of their effects. Our results demonstrate that current FL algorithms are highly susceptible to variations in weight decay values. To reconcile the need to prevent overfitting with the aim to minimize update divergence, we present an adaptive weight decay strategy named *Federated Optimization with Normalized Annealing Regularization* (FedNAR). FedNAR is a cost-efficient approach that can be effortlessly incorporated into any prevailing FL algorithm as an algorithmic plug-in. At its core, we govern the magnitude of each weight update by limiting the norm of the sum of the gradient and weight decay. We conduct a meticulous theoretical analysis for

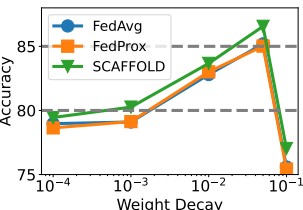

Figure 1: Accuracy of three FL algorithms (*i.e.,* FedAvg, FedProx, and SCAFFOLD) after 1000 rounds using different weight decay coefficients. See details in Section 3.

universally adaptive weight decay functions, proving that our FedNAR is a logically sound choice for achieving convergence from a theoretical perspective. Through an array of experiments spanning computer vision and language tasks, we establish that FedNAR fosters improved convergence and facilitates superior final accuracy.

**Our contributions.**  More specifically, we made the following contributions in this work to the entire federated learning and optimization community

- We empirically observed that FL settings make weight decay more crucial in training.
- Based on the observation described above, we designed FedNAR, an algorithmic plug-in to control the magnitude of weight decay that is compatible with all existing FL algorithms.
- We propose the first theoretical framework of FL that takes weight decay into account and show that FedNAR is guaranteed to converge under FL scenarios.
- We conduct extensive experiments on vision and language tasks by plugging FedNAR into state-of-the-art baselines, which demonstrate that adopting FedNAR generally leads to faster training and higher accuracy.

## 1.1  Related work

FL is a paradigm for collaborative learning with decentralized private data [7, 1, 4, 2, 3]. Standard approach to FL tackles it as a distributed optimization problem where the global objective is defined by a weighted combination of clients' local objectives [8, 4, 9, 10]. Theoretical analysis has demonstrated that federated optimization exhibits convergence guarantees but only under certain conditions, such as a bounded number of local epochs [11]. Other work has tried to improve the averaging-based aggregations [12, 13]. Techniques such as secure aggregation [14, 15, 16] and differential privacy [17, 18] have been widely adopted to further improve privacy in FL [19]. Our FedNAR is based on the standard FedAvg [1] framework and can be adapted to different settings.

In order to enhance convergence in a communication-efficient framework, addressing the challenges posed by highly imbalanced data, we highlight the significance of weight decay, which is crucial in modern deep neural network (DNN) training. Extensive research in centralized training has explored various methods to incorporate weight decay into optimization algorithms, such as AdamW [20] and WIN [21]. However, the integration of weight decay into FL remains relatively unexplored, both in terms of theoretical understanding and practical implementation. A recent study by [22] delved into the examination of weight decay during server updates in the context of FL. Nevertheless, their focus was primarily on tuning the global weight decay parameter, neglecting the exploration of weight decay in local updates. Additionally, their choice of the global weight decay parameter relied on a proxy dataset, which is unrealistic and lacks theoretical guarantees.

## 2  Preliminaries: federated optimization formulation

Our objective is to minimize the loss function $F(x)$ of a model $x$, which is trained on data dispersed across $M$ clients, as denoted by:

$$x^* = \arg\min_x F(x) = \arg\min_x \sum_{i=1}^{M} F_i(x),$$

where $F_i(x)$ corresponds to the loss function calculated using the data located on the $i$-th client. Federated optimization introduces two key constraints: privacy and communication efficiency. The privacy constraint ensures that clients retain their data without sharing, whereas the communication constraint emerges due to the substantial cost and restricted capacity of network bandwidth. As a result, only a fraction of clients can engage in each communication round with the server in practice. To circumvent these constraints, FedAvg has been proposed as a prevalent communication-efficient FL framework [1]. For a round $1 \leq t \leq T$, we show an illustration of FedAvg in Algorithm 1.

---

**Algorithm 1** Round $t$ of FedAvg

---

**Require:** Global model $x^{t-1}$, learning rate $\lambda_l, \lambda_g$, number of local updates $\tau$.
  1: **for** Client $i = 1, \cdots, M$ **do**
  2:    Let $x_0^{t,i} = x^{t-1}$. Update $x_k^{t,i} = x_{k-1}^{t,i} - \lambda_l \nabla F_i(x_{k-1}^{t,i})$ for $k = 1, \cdots, \tau$.
  3:    Send local update $\Delta^{t,i} = x_0^{t,i} - x_\tau^{t,i}$ back to the server.
  4: **end for**
  5: Aggregate local updates: $\Delta^t = \frac{1}{M} \sum_{i=1}^{M} \Delta^{t,i}$, and update global model: $x^t = x^{t-1} - \lambda_g \Delta^t$.

---

In the original FedAvg formulation, local updates on each client are realized through (stochastic) gradient descent (SGD/GD) with the loss function $F_i(x)$, identical to the global objective. On the FedAvg foundation, we can further adjust the local loss function to be $F_i(x) + \zeta_i(x)$, where $\zeta_i(x)$ serves as a regularization term, enabling faster convergence in the federated learning context. For instance, "client drift", a common problem in federated learning due to multiple local updates preceding aggregation, can be mitigated by adding a proximal term. FedProx [4] employs a local loss $F_i(x) + \frac{\mu}{2}\|x - x_0\|^2$ that incorporates an additional proximal term $\zeta_i(x) = \frac{\mu}{2}\|x - x_0\|^2$ to counteract overfitting in local clients. Similarly, SCAFFOLD [6] introduces local control variables $c_i$ and a global control variable $c$ and modifies the local gradient to be $\nabla F_i(x) - c_i + c$. Thus, $\zeta_i(x)$ equates to $(c - c_i)^T x$.

## 3 An empirical study of weight decay

Weight decay is a well-established approach in standard centralized deep neural network training, primarily used to prevent overfitting [23, 20]. In the realm of FL, weight decay values generally mirror those employed in centralized training. In this section, we meticulously explore the impact of weight decay on federated optimization. In addition to the original FedAvg, we investigate various methodologies including {FedProx, SCAFFOLD}, which adjust local updates, and {FedAvgm, FedAdam} [9], which utilize SGD with momentum and Adam, respectively, in the global update while treating aggregated local updates as pseudo-gradient. We also assess FedExP [24], which adaptively modulates the global learning rate.

We evaluate all the aforementioned algorithms on the CIFAR-10 dataset, partitioned among 100 clients under three different settings. Following [25], in each setting, we initially sample a class distribution $\mathbf{p}_i \sim \text{Dirichlet}(\alpha, \ldots, \alpha)$ for each client $i$, and subsequently sample training data in accordance with the distributions $\mathbf{p}_i$. When the value of $\alpha$ is low, the sampled individual probabilities $\mathbf{p}_i$ exhibit higher concentration, whereas for greater values of $\alpha$, the concentration diminishes. Consequently, a larger value of $\alpha$ corresponds to a more balanced data distribution across clients.

We train a ResNet-18 for each algorithm on CIFAR-10 for our empirical study. Figure 2 presents the test accuracy plotted against weight decay values from $\{10^{-4}, 10^{-3}, 10^{-2}, 5 \times 10^{-2}, 10^{-1}\}$ for each setting. The $y$-axis range is maintained consistently between $(50, 95)$ across all settings to facilitate a clear comparative analysis of the influence of weight decay under varied settings.

In the baseline setting (a), the influence of weight decay on the performance of each algorithm is pronounced in two primary ways. Firstly, a noteworthy enhancement in accuracy exceeding $6\%$ is observed when leveraging optimal weight decay as compared to sub-optimal weight decay. This improvement outstrips the differences discerned across various algorithms. Secondly, increasing weight decay to be slightly larger than the optimal value, such as from 0.05 to 0.1 or from 0.01 to 0.05, can cause a considerable plunge in accuracy exceeding $10\%$ and even verging on $30\%$. This underscores the phenomenon of amplified update deviation in FL. However, both of these elements are tempered when either the number of local updates is curtailed in (b) or a balanced data distribution

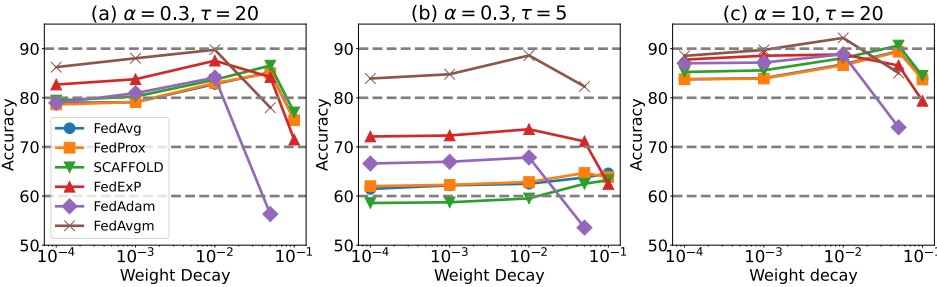

Figure 2: Influence of weight decay for different settings. We train each algorithm (*i.e.,* FedAvg, FedProx, SCAFFOLD, FedExp, FedAdam, FedAvgm) over 1000 rounds with $\tau$ local steps per round, a local learning rate of 0.01, and a global learning rate of 1.0. We apply a decay for the local learning rate of 0.998 per round and a gradient clipping of max norm 10 as per [24]. Given that multiple local steps and imbalanced data distribution are two distinguishing features of FL, we utilize various pairs of $(\alpha, \tau)$ to observe their influence on the results. The baseline setting is chosen to be $(\alpha, \tau) = (0.3, 20)$, resulting in a highly imbalanced data distribution. The second configuration reduces the number of local updates to $(\alpha, \tau) = (0.3, 5)$. The third configuration employs a balanced data distribution with $(\alpha, \tau) = (10, 20)$.

is deployed in (c). In these scenarios, the accuracy curve corresponding to weight decay is more leveled for each algorithm, and the dip in accuracy is significantly reduced when weight decay is slightly elevated from the optimal value.

This finding underscores how the training paradigm in FL can modulate the impact of weight decay. However, prior work has largely overlooked the role of weight decay, both theoretically and experimentally. Convergence analyses in prior studies did not take into account the influence of weight decay and assigned its value as 0. Moreover, previous experiments either adopted values akin to those utilized in centralized training, or fine-tuned it within a restricted range, or simply set it to 0.

## 4 FedNAR and its convergence analysis

Driven by the recognized shortcomings of previous studies, we introduce the first analytical framework for convergence that incorporates local weight decay, examining its resulting deviation and how it impacts global updates. Following this theoretical exploration, we propose our FedNAR algorithm. Designed with the dual aims of ensuring a theoretical guarantee and achieving an optimal rate of convergence, FedNAR marks a significant advancement in this field.

### 4.1 A general federated optimization framework

Our theoretical analysis can cover adaptive learning rate and weight decay. Say the model is of dimension $d$. We first define two functions $\lambda, \mu : \mathbb{N} \times \mathbb{R}^d \times \mathbb{R}^d \to \mathbb{R}^+$, and for $t \geq 0$, we denote $\lambda_t = \lambda(t, \cdot, \cdot), \mu_t = \mu(t, \cdot, \cdot)$ as mappings with the domain of definition on the gradient space and weight space. These two functions stand for learning rate and weight decay respectively. This framework can be adapted to any FL algorithms by making the learning rate and weight decay to be adaptive. We present the framework in Algorithm 2. This framework holds for arbitrary functions $\lambda$ and $\mu$. We will clarify the choices of $\lambda$ and $\mu$ in FedNAR later in equation 3 and 4 after some theoretical analysis, and make clear where the "Normalized Annealing Regularization" in FedNAR comes from. In the following, we denote $f_k^{t,i}(x)$ to be the loss function computed on a stochastic batch instead of $F_i(x)$.

Algorithm 2 is a generalized version of FedAvg that incorporates adaptive local learning rate and adaptive weight decay. We can recover the original FedAvg by setting $\lambda_t(g, x)$ and $\mu_t(g, x)$ to be constant functions. It is worth emphasizing that although most previous works like FedAvg and FedProx employed local weight decay, they did not integrate it into their theoretical analysis. However, our empirical investigations in Section 3 demonstrate that weight decay could significantly affect the convergence of federated optimization. Therefore, we argue that thorough empirical and theoretical analysis is needed to better understand this term.

### 4.2 Convergence analysis and FedNAR

To initiate the theoretical analysis, we present three standard assumptions that are widely used in federated optimization literature [24, 10]. The initial two are conventional assumptions utilized in the general analysis of SGD optimizer, while the third one is common in FL theory.

**Assumption 1** (Smooth loss functions). *Every local loss function is $L$-smooth, i.e., there exists $L > 0$ such that for any client $1 \leq i \leq M$, the gradient satisfies $\|\nabla F_i(x) - \nabla F_i(y)\| \leq L\|x - y\|$.*

---

**Algorithm 2** FedNAR

**Require:** Functions $\lambda_t, \mu_t, F_i$, constant $\lambda_g, T, \tau > 0$, sets $C_t$.
1: **for** Communication rounds $t = 1, \cdots, T$ **do**
2:     Global server sends model $x^{t-1}$ to participated clients in round $t$.
3:     **for** Client $i \in C_t$ **do**
4:         Let $x_0^{t,i} = x^{t-1}$.
5:         **for** Iteration $k = 1, \cdots, \tau$ **do**
6:             Compute (stochastic) gradient: $g_{k-1}^{t,i} = \nabla f_{k-1}^{t,i}(x_{k-1}^{t,i})$.
7:             Compute adaptive learning rate and weight decay:

$$\lambda_{k-1}^{t,i} = \lambda_t(g_{k-1}^{t,i}, x_{k-1}^{t,i}), \quad \mu_{k-1}^{t,i} = \mu_t(g_{k-1}^{t,i}, x_{k-1}^{t,i}).$$

8:             Update $x_k^{t,i} = (1 - \mu_{k-1}^{t,i})x_{k-1}^{t,i} - \lambda_{k-1}^{t,i}g_{k-1}^{t,i}$.
9:         **end for**
10:         Send local update $\Delta^{t,i} = x_0^{t,i} - x_\tau^{t,i}$ back to the server.
11:     **end for**
12:     Aggregate local updates: $\Delta^t = \frac{1}{|C_t|}\sum_{i=1}^{|C_t|}\Delta^{t,i}$, and update global model: $x^t = x^{t-1} - \lambda_g\Delta^t$.
13: **end for**

---

**Assumption 2** (Unbiased gradient and bounded variance). *For any client $1 \leq i \leq M$, the stochastic gradient is an unbiased estimator of the full-batch gradient, i.e., $\mathbb{E}[\nabla f_k^{t,i}(x)] = \nabla F_i(x)$, and there exists $\sigma^2 > 0$, such that for any $1 \leq i \leq M$, $\mathbb{E}[\|\nabla f_k^{t,i}(x) - \nabla F_i(x)\|^2] \leq \sigma^2$.*

**Assumption 3** (Bounded heterogeneity). *There exists $\sigma_g^2 > 0$, such that $\frac{1}{M}\sum_{i=1}^M \|\nabla F_i(x) - \nabla F(x)\|^2 \leq \sigma_g^2$ holds for any $x \in \mathbb{R}^d$.*

Our findings suggest that certain limitations on $\lambda_t$ and $\mu_t$ are necessary to ensure the convergence of FedNAR. We begin by presenting a lemma that demonstrates the impact of local adaptive weight decay and gradients on the global update process.[1]

**Lemma 1.** *For round $t \geq 1$, the global update is equivalent to be*

$$x^t = (1 - \mu_g^{t-1})x^{t-1} - \lambda_g h^t,$$

*where*

$$\mu_g^{t-1} := \lambda_g - \frac{\lambda_g}{M}\sum_{i=1}^M\prod_{j=0}^{\tau-1}(1 - \mu_j^{t,i}), \quad h^t := \frac{1}{M}\sum_{i=1}^M\sum_{j=0}^{\tau-1}\lambda_j^{t,i}\prod_{r=j}^{\tau-1}(1 - \mu_r^{t,i})g_j^{t,i}. \quad (1)$$

We show the full proof in Appendix B. Lemma 1 highlights that the additional weight decay in local updates introduces a deviation $\mu_g^{t-1}x^{t-1}$ in the global update. Previous works on federated optimization [4, 24, 6] did not include weight decay in their theoretical analysis, leading to $\mu_j^{t,i}$ always being zero, and $\mu_g^{t-1} = \lambda_g - \frac{\lambda_g}{M} \cdot M = 0$. Moreover, we can see that larger $\tau$ further intensifies the deviation, since $\mu_g^{t-1} = \lambda_g(1 - \frac{1}{M}\sum_{i=1}^M\prod_{j=0}^{\tau-1}(1 - \mu_j^{t,i}))$, and larger $\tau$ implies that $\mu_g^{t-1}$ increases and approaches $\lambda_g$. This highlights that the communication-efficient training framework of FL amplifies the impact of weight decay.

So to achieve convergence with local weight decay, it is necessary to regulate this deviation $\mu_g^t x^t$. Therefore, we present the following result to control the norm of the global model, *i.e.*, $\|x^t\|$, as the first step.

**Theorem 1.** *If the functions $\lambda_t$ and $\mu_t$ satisfy that there exists $A > 0$, such that for any $t \geq 0$, and $(g, x) \in \mathbb{R}^d \times \mathbb{R}^d$, $\|\lambda_t(g,x)g + \mu_t(g,x)x\| \leq A$, then the norm of global model in Algorithm 2 is at most polynomial increasing with respect to $t$, i.e., $\|x_t\| \leq \mathcal{O}(poly(t))$.*

We want to emphasize that the condition required for $\lambda$ and $\mu$ to achieve polynomial increasing speed in Theorem 1 is practical and commonly used in real-world training. Many previous works, such as

---

[1]Here we assume full-client participation, following previous works [10, 24]. However, our analysis can be easily extended to partial-client participation, as demonstrated in [10].

[26, 24], adopt gradient scaling based on norm, which bounds the term $\|\lambda_t(g,x)g\|$ by a constant related to the clipping value. Similarly, we can bound $\|\mu_t(g,x)x\|$ using norm clipping, for example, by setting $\mu_t(g,x) = \mu \min\{1, c/\|x\|\}$ for some constant $c$ and $\mu$. Therefore, the condition is easily satisfied from a practical standpoint.

Finally, we give the convergence result for FedNAR in the following Theorem.

**Theorem 2.** *Under Assumptions 1, 2 and 3, if $\lambda_t$ and $\mu_t$ satisfy the condition in Theorem 1, and also if there exists $\{u_t\}$ with an exponential decay speed, such that for any $t \geq 0$, and any $g, x \in \mathbb{R}^d$, $\mu_t(g,x) \leq u_t$, then the global model $\{x^t\}$ satisfies:*

$$\min_{1 \leq t \leq T} \mathbb{E}\|\nabla F(x^t)\|^2 \leq \mathcal{O}\left(\frac{1}{T}\right) + \mathcal{O}\left(\underbrace{L\lambda_g^2\tau^2 C^2}_{\text{weight decay error}} + \underbrace{L\lambda_g^2\tau l_*^2(\tau\sigma^2 + L^2\tau^2 A^3 + \sigma_g^2)}_{\text{client drift and data heterogeneity error}}\right),$$

*where $l_* = \max_t\{l_t\}$, with $l_t = \max_{g,x}\{\lambda_t(g,x)\}$, and $C$ satisfies $u_t\|x^t\|, u_t\|x^t\|^2 \leq C$ for any $t$.*

**Discussions.** For the first time, we demonstrate that federated optimization, with non-convex objective functions and stochastic local gradient, achieves convergence when employing adaptive learning rate and weight decay. Our findings resemble previous studies [26, 10], featuring a diminishing term $\mathcal{O}(1/T)$ as well as a non-vanishing term arising from weight decay, client drift, and data heterogeneity in FL. Notably, our bound recovers previous results when weight decay is disregarded, *i.e.*, $C = 0$.

The detailed proof can be found in the Appendix B. Here we briefly explain the condition of exponential decreasing. To tackle the case with deviation caused by local weight decay, one of the key steps is to bound $\|\mu_g^t x^t\|$ by a constant as shown in the complete proof. So for this term, we have

$$\|\mu_g^t x^t\| = \lambda_g\left(1 - \frac{1}{M}\sum_{i=1}^M \prod_{j=0}^{\tau-1}\left(1 - \mu_j^{t+1,i}\right)\right)\|x^t\| \leq \lambda_g\left(1 - \frac{1}{M}\sum_{i=1}^M (1 - u_{t+1})^\tau\right)\|x^t\|$$

$$\leq \lambda_g\left(1 - \frac{1}{M}\sum_{i=1}^M (1 - \tau u_{t+1})\right)\|x^t\| = \lambda_g\tau u_{t+1}\|x^t\| \leq \lambda_g\tau C, \tag{2}$$

which is a constant unrelated to $t$. Here the second inequality is from $(1-\alpha)^n \geq 1 - n\alpha$ for $0 \leq \alpha \leq 1$ and $n \geq 1$. We can see that the exponential decrease condition can be released as long as the decay of $\{u_t\}$ can control $\{\|x^t\|\}$. Since we have already shown a polynomial increasing rate for $\{\|x^t\|\}$, a convenient choice for both theory and practice is to set $\{u_t\}$ an exponential decrease. We summarize the conditions of $\lambda$ and $\mu$ needed to provide convergence below:

**Condition 1.** *There exists $A > 0$, such that for any $t \geq 0$ and $(g,x) \in \mathbb{R}^d \times \mathbb{R}^d$, $\|\lambda_t(g,x)g + \mu_t(g,x)x\| \leq A$. There exists $\{u_t\} \downarrow 0$ with exponential decay, such that for any $t \geq 0$, and any $(g,x) \in \mathbb{R}^d \times \mathbb{R}^d$, $\mu_t(g,x) \leq u_t$.*

**Choice of $\lambda$ and $\mu$.** In order to meet the requirement of Condition 1, a straightforward and intuitive way is to use clipping. We start by selecting two sequences $l_t$ and $u_t$, where $u_t = u_0\gamma^t$ for some $\gamma < 1$ is the control sequence of $\mu_t$. The sequence $\{l_t\}$ is the learning rate schedule, and is usually set to be a constant or decrease over training. Then we define

$$\lambda_t(g,x) := \begin{cases} l_t \cdot A/\|g + u_t x/l_t\|, & \text{if } \|g + u_t x/l_t\| > A \\ l_t, & \text{otherwise} \end{cases}, \tag{3}$$

and

$$\mu_t(g,x) := \begin{cases} u_t \cdot A/\|g + u_t x/l_t\|, & \text{if } \|g + u_t x/l_t\| > A \\ u_t, & \text{otherwise} \end{cases}. \tag{4}$$

We can easily verify that the choices of $\lambda_t$ and $\mu_t$ given in equations 3 and 4 satisfy Condition 1. See details in Appendix B. Hence, we choose these functions as the input of Algorithm 2.

### 4.3 Understanding FedNAR

**Normalized annealing regularization.** Now we can explain why we call our method "Normalized Annealing Regularization". First, the regularization term is co-normalized with the gradient term.

Second, the "annealing" comes from two aspects: (1) The sequence $u_t$ decays with an exponential rate, and $\mu_t(g, x) \leq u_t$ uniformly for all $(g, x)$. (2) We observe from experiments that the norm of $x$ keeps increasing, so the term $\|g + u_t x/l_t\|$ will also keep increasing. NAR computes $A/\|g + u_t x/l_t\|$ which by the preceding argument is a decreasing term, thus introducing another form of decay. We will provide further illustrations of this in the experiment section.

**Flexibility of FedNAR.** Since FedNAR's convergence analysis does not impose any extra conditions on the local loss functions, it generalizes several FL algorithms with distinct local training objectives, while conferring upon them theoretical assurances comparable to FedNAR. For example, FedProx and SCAFFOLD only differ from FedAvg on local loss functions. Thus, we only need to change Line 6 in Algorithm 2 to the corresponding loss functions in FedProx and SCAFFOLD to obtain FedProx-NAR and SCAFFOLD-NAR.

**Comparison with gradient clipping.** Gradient clipping [27] is a widely used technique in federated optimization to enhance training stability as demonstrated by various studies [26, 24, 28]. It is achieved by constraining the norm of gradients in each iteration. The technique involves setting a threshold value and rescaling the gradients if they exceed this limit. In our framework, gradient clipping can be represented by setting

$$\lambda_t(g, x) := \begin{cases} l_t \cdot A/\|g\|, & \text{if } \|g\| > A \\ l_t, & \text{otherwise} \end{cases}, \text{ and } \quad \mu_t(g, x) := u_t. \tag{5}$$

Our proposed FedNAR distinguishes itself from previous gradient clipping strategies in two primary ways. First, both our weight decay and learning rate functions are adaptive, with dependencies on the current gradient and model weights. This contrasts with traditional gradient clipping, which only adapts the learning rate function based on the current gradient. Second, instead of solely employing the norm of the gradient, FedNAR adopts the norm of the sum of the gradient and weight decay as the clipping criterion. This strategy ensures that each local update is bounded, thereby preventing potential explosions caused by large decay terms $\mu_k^{t,i} x_k^{t,i}$ in federated optimization. Consequently, FedNAR not only guarantees convergence but also bolsters training stability within the context of federated optimization.

**Implementation of FedNAR.** Implementing FedNAR merely requires specifying an initial weight decay and a decay rate. Therefore, for state-of-the-art baseline methods that employ gradient clipping [24, 26], no additional hyperparameters are needed. Fundamentally, FedNAR can be interpreted as a modest alteration that adjusts the sequence of gradient clipping and weight decay operations. Consequently, the implementation of FedNAR can be as uncomplicated as modifying a single line of code, resulting in a solution that is both highly effective and efficient.

## 5 Experiments

We show the results of FedNAR for both vision and language datasets in this section.

### 5.1 Main results

**Experiments on the CIFAR-10 dataset.** As outlined in Section 3, we manually partition the CIFAR-10 dataset into 100 clients, following the methodology described in [25, 24]. Specifically, we randomly assign a distribution $\mathbf{p}_i \sim \text{Dirichlet}(\alpha, \ldots, \alpha)$ to each client $i$, then draw data from the training set in line with these distributions $\{\mathbf{p}_i\}$. We choose $\alpha$ from the set $\{0.3, 1, 10\}$, with larger $\alpha$ values indicating a more balanced data distribution across each client. With $\alpha = 0.3$, the local data distribution is highly skewed, whereas with $\alpha = 10$, the local distribution is nearly balanced across all classes. We adhere to the training parameters and settings, optimally tuned as per [24]. For each algorithm, we conduct 1000 rounds of training for full convergence, with 20 steps of local training per round. In each round, we randomly sample 20 clients. For the baseline algorithms, we apply gradient clipping as suggested in [24, 26]. We set the local learning rate to 0.01 with a decay of 0.998, and cap the maximum norm at 10 for both gradient clipping and FedNAR to ensure a fair comparison.

Table 1 showcases the performance of FedAvg-NAR, FedProx-NAR, and SCAFFOLD-NAR across different $\alpha$ values. The results reveal a consistent performance boost offered by FedNAR across a variety of $\alpha$ values, with particularly significant improvements observed for skewed distributions. Test accuracy after each training round is also illustrated in Figure 3.

The backbone algorithms, FedProx and SCAFFOLD, are variants of FedAvg that modify the local training objective. FedNAR can be tailored to these client-side FedAvg variants, which provide a

Table 1: Experimental results on the CIFAR-10 dataset, where the FedNAR plugin is incorporated into the FedAvg, FedProx, and SCAFFOLD algorithms, with data partitioning across varying levels of heterogeneity controlled by $\alpha$.

| Algorithm | $\alpha = 0.3$ | | $\alpha = 1$ | | $\alpha = 10$ | |
| --- | --- | --- | --- | --- | --- | --- |
| | Baseline | FedNAR | Baseline | FedNAR | Baseline | FedNAR |
| FedAvg | 85.19 | **87.64** | 88.50 | **89.55** | 89.45 | **90.23** |
| FedProx | 85.02 | **87.45** | 88.56 | **89.58** | 89.37 | **90.38** |
| SCAFFOLD | 86.89 | **89.17** | 89.52 | **91.13** | 90.64 | **91.85** |
| FedExP | 86.46 | **88.55** | 88.28 | **89.51** | 88.82 | **89.69** |
| FedAdam | 84.12 | **85.81** | 87.35 | **87.78** | 88.86 | **89.56** |
| FedAvgm | 90.33 | **90.43** | 91.60 | **91.76** | **92.19** | 91.15 |

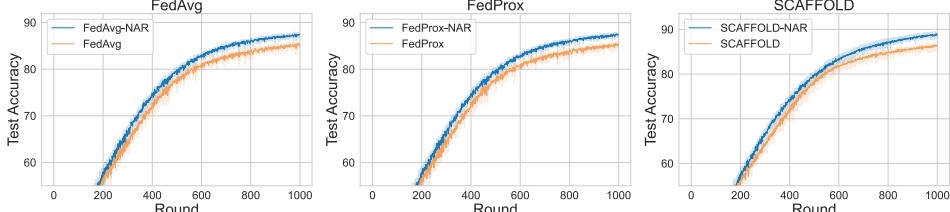

Figure 3: Test accuracy curve for FedAvg, FedProx, SCAFFOLD and their FedNAR variants for $\alpha = 0.3$. For each training, we take 3 random seeds.

guarantee of convergence. Moreover, FedNAR can also be incorporated into server-side FedAvg variants, which retain the local training but alter the global update. Table 1 showcases the results of these implementations, using the same hyperparameter configurations. These results corroborate the same conclusion.

**Experiments on the Shakespeare dataset.** The Shakespeare dataset [29], derived from *The Complete Works of William Shakespeare*, assigns each speaking role in every play to a unique client, leading to an inherent *non-i.i.d.* partition. We select 100 clients and train a model for the task of next-character prediction, incorporating 80 possible characters, following previous studies [29, 26]. We train a transformer model with 6 attention layers for feature extraction alongside a fully connected layer for predictions.

Table 2: Experimental results on the Shakespeare dataset, where the FedNAR plugin is incorporated into the FedAvg, FedProx, and SCAFFOLD algorithms. WD denotes the initial weight decay value applied in the first round.

| Algorithm | WD = $10^{-4}$ | | WD = $10^{-3}$ | |
| --- | --- | --- | --- | --- |
| | Baseline | FedNAR | Baseline | FedNAR |
| FedAvg | 42.44 | **44.37** | 40.05 | **44.69** |
| FedProx | 41.92 | **43.57** | **40.91** | 40.26 |
| SCAFFOLD | 47.55 | 45.60 | 42.40 | **44.53** |
| FedExP | 38.15 | **39.63** | 39.67 | **39.92** |
| FedAdam | **45.80** | 45.42 | 47.19 | **47.48** |
| FedAvgm | 45.03 | **45.63** | 46.12 | **47.13** |

Each character is represented by a 128-dimensional vector, and the transformer's hidden dimension is set to 512. The training regimen spans 300 rounds, with 20 clients chosen at random in each round and 5 local epochs executed on each client. During local training, we utilize a batch size of 100, a learning rate of 0.1 with a decay rate of 0.998 per round, a dropout rate of 0.1, and a maximum norm of 10 for both the baseline algorithms and FedNAR. Global updates are performed with a learning rate of 1.0. As in the previous experiment, FedNAR is integrated into six core algorithms. We also employ two distinct weight decay values to showcase FedNAR works for different hyperparameters. The results are presented in Table 2.

## 5.2 Ablation studies

In this section, we conduct some ablation studies. For all the experiments, we run FedAvg on CIFAR-10 with 100 clients and $\alpha = 0.3$ for 1000 rounds, where each round consists of updates from 20 clients and 20 steps on each client. The local learning rate is 0.01 with a decay of 0.998.

**FedNAR's robustness in relation to hyperparameters.** As demonstrated in Section 3, weight decay exhibits high sensitivity in federated optimization and requires meticulous tuning. However, due to its co-normalization mechanism, FedNAR has demonstrated a greater robustness towards the selection of the initial weight decay. Employing the same experimental framework as in Section 3, we observed that initiating weight decay at 0.1 in FedAvg significantly compromised model performance,

inciting a gradient explosion due to an overly high weight decay. This led to a performance standard similar to a model with no weight decay. Yet, FedNAR was found to navigate this hurdle through self-adjustment, leading to considerable performance enhancements during the latter stages of training, in spite of the initial detrimental impact on performance. Additionally, further experiments on FedProx and SCAFFOLD also manifested a similar occurrence. The test accuracy curve is depicted in Figure 4.

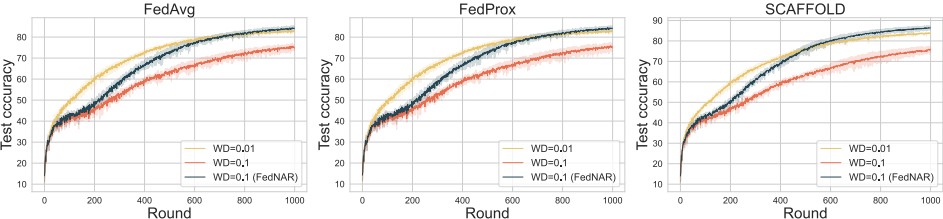

Figure 4: The self-adjusting capability of FedNAR. WD denotes the initial weight decay value applied in the first round. It is crucial to note that in each round $t$, the weight decay remains consistent for both the baseline methods and FedNAR. The distinction lies in FedNAR's adoption of co-clipping across both weight decay and gradient. We drew comparisons among FedAvg, FedProx, and SCAFFOLD with WD values of 0.1 and 0.01. The utilization of 0.1 as the WD value proved to be less than ideal, resulting in a performance decrement in the baseline methods. Conversely, FedNAR initially mimics this trend but swiftly ameliorates during the subsequent stages, outstripping the baselines even with a more favorable initial WD value of 0.01.

**Frequency and strength of clipping in FedNAR.** In Section 4.3, we showcased how FedNAR employs an "annealing" process to stave off excessive deviation precipitated by weight decay, from two angles. The first angle pertains to the exponential decay in $u_t$, while the second involves co-clipping with the gradient. To delve deeper into the frequency and intensity of the clipping operation, we scrutinized the updates that were subjected to clipping. In particular, we evaluated the count of clipped updates and the average norm of the update $\|g + \frac{u_t}{l_t}x\|_2$, signifying the intensity of clipping, for these steps, as depicted in Figure 5. Our analysis points out that both the count and the norm of updates experiencing clipping escalate over time. This observation implies that the co-clipping term gains increasing prominence during the training process, aiding in the annealing of weight decay which is proportional to $1/\|g + \frac{u_t}{l_t}x\|_2$.

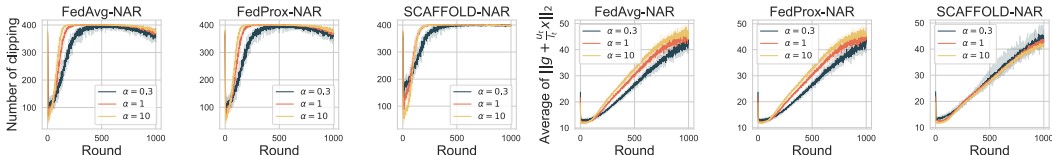

Figure 5: Frequency and strength of clipping. In every round, there is a sum total of 20 clients, and each client carries out 20 updates, leading to a collective tally of 400 updates per round. We monitor the count of clipping instances within these 400 updates during each round and compute the average norm of the updates subjected to clipping. We execute experiments for diverse $\alpha$ values, and for every algorithm, we present the outcomes utilizing three distinct seeds.

## 6  Limitation

As an inaugural exploration into federated optimization with weight decay, we concentrated on examining the theoretical attributes of FL algorithms that utilize gradient descent updates both server-side and client-side. Future investigations can extend this to encompass different server update methodologies. Furthermore, due to privacy restrictions, we could not carry out experiments using real heterogeneous data, such as hospital data. Instead, we relied on simulated data distributions or naturally non-*i.i.d* splits, following the conventional methods employed in prior FL algorithm research [1, 26, 10, 24, 3, 29].

## 7  Conclusion

In this study, we delve into the influence of weight decay in FL, underscoring its paramount importance in traditional FL scenarios characterized by multiple local updates and skewed data distribution. To explore these facets, we present an innovative theoretical framework for FL that integrates weight decay into convergence analysis. Upon discerning the conditions for convergence, we introduce FedNAR, a solution that conforms to these prerequisites. FedNAR boasts a straightforward implementation and can be effortlessly adapted to a variety of backbone FL algorithms. In our experiments on vision and language datasets, we consistently record significant performance improvements across

all backbone FL algorithms when we employ FedNAR, leading to accelerated convergence rates. In addition, our ablation studies illustrate FedNAR's heightened robustness against hyperparameters. Going forward, our aim is to expand this theoretical framework to include a wider array of FL algorithms, incorporating adaptive optimization methods.

## Acknowledgments

We thank the anonymous reviewers for their valuable insights and recommendations, which have greatly improved our work. This research has been graciously funded by NGA HM04762010002, NSF IIS1955532, NSF CNS2008248, NIGMS R01GM140467, NSF IIS2123952, NSF BCS2040381, an Amazon Research Award, NSF IIS2311990, and DARPA ECOLE HR00112390063.

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
