# Contents of the Appendix

# A  Additional experimental results

In this section, we present additional accuracy curves from our experiments, highlighting the superior performance and faster convergence of FedNAR in almost all cases.

## A.1  CIFAR-10 dataset

Figure 6 displays test accuracy curves for all six backbone algorithms under three distinct imbalance parameters: $\alpha \in \{0.3, 1, 10\}$. The results clearly demonstrate that FedNAR outperforms the baselines, particularly in scenarios with imbalanced data.

## A.2  Shakespeare dataset

The experimental results presented in Figure 7 and 8 showcase the outcomes of experiments performed on the Shakespeare dataset. Six backbone algorithms were utilized, with initial weight decay values selected from $\{10^{-3}, 10^{-4}\}$. These findings serve as evidence that FedNAR, as an adaptive weight decay scheduling algorithm, exhibits effectiveness across various initial weight decay values.

# B  Supplementary proofs

In this section, we provide a comprehensive proof for Lemma 1, Theorem 1, and Theorem 2, which are discussed in Section B.1 and Section B.2. These proofs pertain to the general framework 2, incorporating adaptive learning rate and weight decay. Additionally, in Section B.3, we focus on the specific context of FedNAR and present a detailed analysis of its properties by showcasing interesting characteristics.

## B.1  Proof of Lemma 1 and Theorem 1

These two results exemplify the distinctive update dynamics exhibited by our framework, encompassing both the specific update formula and an upper bound constraint on the parameter norm.

**Lemma 1.** *For round $t \geq 1$, the global update is equivalent to be*

$$x^t = (1 - \mu_g^{t-1})x^{t-1} - \lambda_g h^t,$$

*where*

$$\mu_g^{t-1} := \lambda_g - \frac{\lambda_g}{M} \sum_{i=1}^{M} \prod_{j=0}^{\tau-1}(1-\mu_j^{t,i}), \quad h^t := \frac{1}{M} \sum_{i=1}^{M} \sum_{j=0}^{\tau-1} \lambda_j^{t,i} \prod_{r=j}^{\tau-1}(1-\mu_r^{t,i})g_j^{t,i}. \tag{6}$$

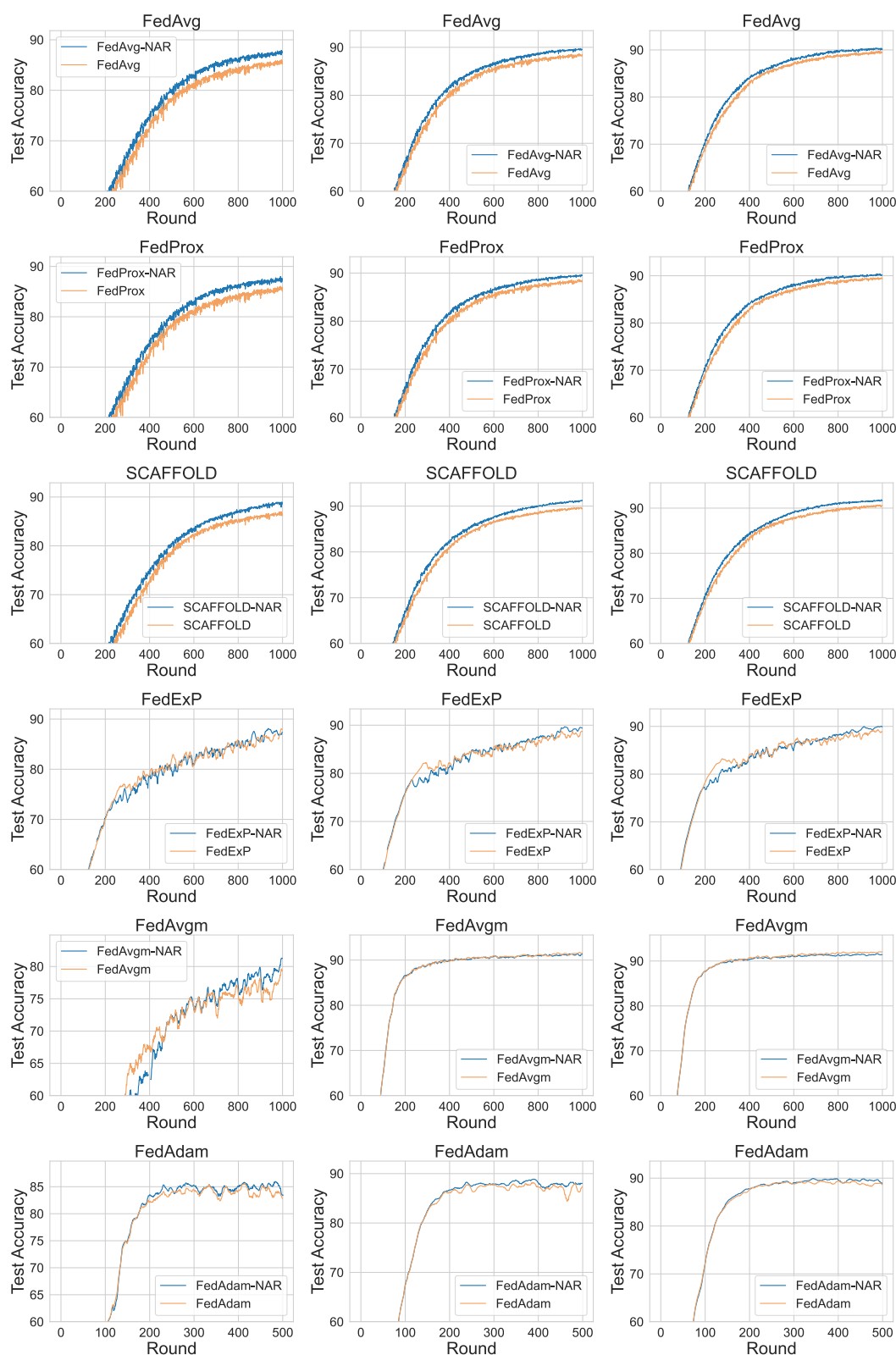

Figure 6: Test accuracy curve for CIFAR-10 dataset for 6 different algorithms. For each algorithm, three columns correspond to the results of $\alpha \in \{0.3, 1, 10\}$ respectively.

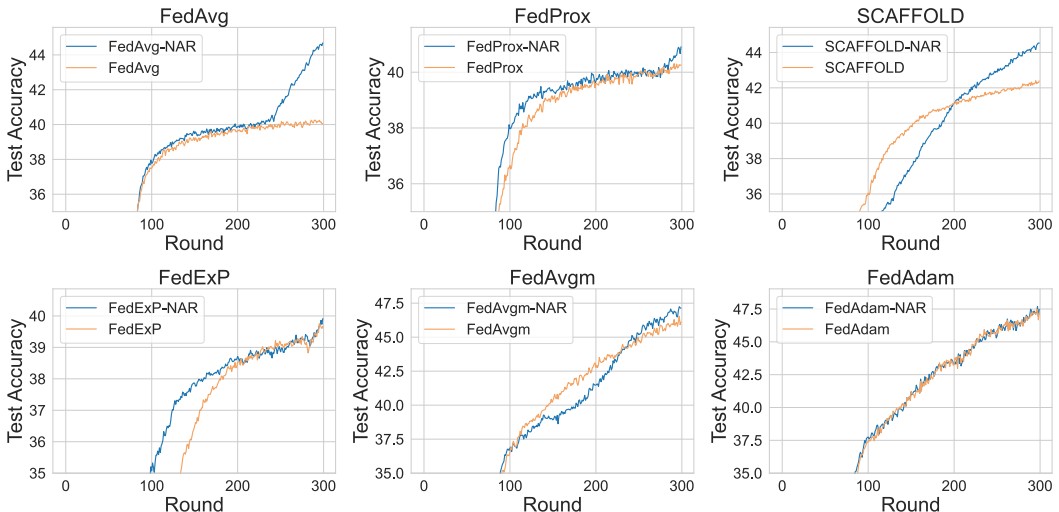

Figure 7: Test accuracy for Shakespeare dataset using different backbone algorithms with initial weight decay $10^{-3}$.

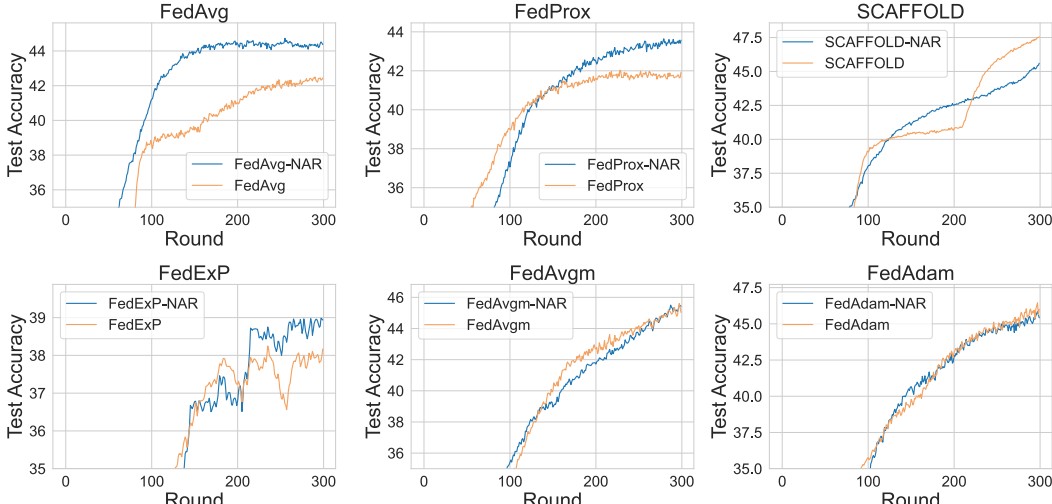

Figure 8: Test accuracy for Shakespeare dataset using different backbone algorithms with initial weight decay $10^{-4}$.

*Proof.* For round $1 \leq t \leq T$ and client $1 \leq i \leq M$, write the the update for iteration $1 \leq k \leq \tau$, we have:

$$x_1^{t,i} = (1 - \mu_0^{t,i})x_0^{t,i} - \lambda_0^{t,i}g_0^{t,i},$$
$$\vdots,$$
$$x_\tau^{t,i} = (1 - \mu_{\tau-1}^{t,i})x_{\tau-1}^{t,i} - \lambda_{\tau-1}^{t,i}g_{\tau-1}^{t,i}.$$

Compute $x_\tau^{t,i}$ iteratively, we have:

$$x_\tau^{t,i} = \prod_{j=0}^{\tau-1}(1 - \mu_j^{t,i})x_0^{t,i} - \sum_{j=0}^{\tau-1}\lambda_j^{t,i}\prod_{r=j}^{\tau-1}(1 - \mu_r^{t,i})g_j^{t,i},$$

local update

$$\Delta^{t,i} = x_0^{t,i} - x_\tau^{t,i} = \left(1 - \prod_{j=0}^{\tau-1}(1 - \mu_j^{t,i})\right) x_0^{t,i} + \sum_{j=0}^{\tau-1} \lambda_j^{t,i} \prod_{r=j}^{\tau-1}(1 - \mu_r^{t,i}) g_j^{t,i},$$

and aggregated local updates (pseudo gradient)

$$\Delta^t = \frac{1}{M} \sum_{i=1}^{M} \Delta^{t,i} = \left(1 - \frac{1}{M} \prod_{j=0}^{\tau-1}(1 - \mu_j^{t,i})\right) x_0^{t,i} + \frac{1}{M} \sum_{i=1}^{M} \sum_{j=0}^{\tau-1} \lambda_j^{t,i} \prod_{r=j}^{\tau-1}(1 - \mu_r^{t,i}) g_j^{t,i}.$$

So the global update is

$$x^t = x^{t-1} - \lambda_g \Delta^t = \left(1 - \lambda_g + \frac{\lambda_g}{M} \sum_{i=1}^{M} \prod_{j=0}^{\tau-1}(1 - \mu_j^{t,i})\right) x^{t-1} - \frac{\lambda_g}{M} \sum_{i=1}^{M} \sum_{j=0}^{\tau-1} \lambda_j^{t,i} \prod_{r=j}^{\tau-1}(1 - \mu_r^{t,i}) g_j^{t,i},$$

which is the form in the Lemma. $\square$

In the following, we denote $[N] := \{1, 2, \cdots, N\}$, and $\xi_t(g, x) := \lambda_t(g, x)g + \mu_t(g, x)x$ for $1 \leq t \leq T$.

**Theorem 1.** *If the functions $\lambda_t$ and $\mu_t$ satisfy that there exists $A > 0$, such that for any $t \geq 0$, and $(g, x) \in \mathbb{R}^d \times \mathbb{R}^d$, $\|\lambda_t(g, x)g + \mu_t(g, x)x\| \leq A$, then the norm of global model in Algorithm 2 is at most polynomial increasing with respect to $t$, i.e., $\|x_t\| \leq \mathcal{O}(poly(t))$.*

*Proof.* For each $(t, i, k) \in [T] \times [M] \times [\tau]$, we have

$$x_k^{t,i} = (1 - \mu_{k-1}^{t,i})x_{k-1}^{t,i} - \lambda_{k-1}^{t,i} g_{k-1}^{t,i} = x_{k-1}^{t,i} - \xi_t(g_{k-1}^{t,i}, x_{k-1}^{t,i}).$$

Therefore,

$$\Delta^{t,i} = x_0^{t,i} - x_\tau^{t,i} = \sum_{k=1}^{\tau} \xi_t(g_{k-1}^{t,i}, x_{k-1}^{t,i}),$$

and

$$\Delta^t = \frac{1}{M} \sum_{i=1}^{M} \Delta^{t,i} = \frac{1}{M} \sum_{i=1}^{M} \sum_{k=1}^{\tau} \xi_t(g_{k-1}^{t,i}, x_{k-1}^{t,i}),$$

also

$$x^t = x^{t-1} - \lambda_g \Delta^t = \cdots = x^0 - \lambda_g \sum_{p=1}^{t} \Delta^p = x^0 - \lambda_g \sum_{p=1}^{t} \frac{1}{M} \sum_{i=1}^{M} \sum_{k=1}^{\tau} \xi_p(g_{k-1}^{p,i}, x_{k-1}^{p,i}).$$

This means

$$\|x^t\| \leq \|x^0\| + \frac{\lambda_g}{M} \sum_{p=1}^{t} \sum_{i=1}^{M} \sum_{k=1}^{\tau} \|\xi_p(g_{k-1}^{p,i}, x_{k-1}^{p,i})\| \leq \|x^0\| + \lambda_g \tau A t,$$

which is linear to $t$, so also a polynomial to $t$. $\square$

### B.2 Proof of Theorem 2

To prove Theorem 2, we begin by introducing a lemma that aims to limit the discrepancy between local updates and one-step gradients. This lemma is a crucial step in the conventional approach observed in theoretical analyses of FL [24, 10]. However, in our case, this process can be considerably simplified due to the bounded nature of our local updates.

### B.2.1 Gap between averaged multi-step local updates and one-step gradient

**Lemma 2.** *For $(t,i) \in [T] \times [M]$, denote*

$$\beta_j^{t,i} = \lambda_j^{t,i} \prod_{r=j}^{\tau-1} (1 - \mu_r^{t,i}), \qquad \beta^{t,i} = \sum_{j=0}^{\tau-1} \beta_j^{t,i}, \quad and \quad h^{t,i} = \frac{1}{\beta^{t,i}} \sum_{j=0}^{\tau-1} \beta_j^{t,i} g_j^{t,i},$$

*which is the accumulated updates in client $i$ and round $t$. Then there exists constant $D > 0$ such that for any $t \in [T]$, we have*

$$\frac{1}{M} \sum_{i=1}^{M} \mathbb{E} \|h^{t,i} - \nabla F_i(x^{t-1})\|^2 \leq D,$$

*where the expectation is with respect to random batches given $x^{t-1}$.*

*Proof.* By definition, we have

$$\mathbb{E} \|h^{t,i} - \nabla F_i(x^{t-1})\|^2 = 2\mathbb{E} \left\| \frac{1}{\beta^{t,i}} \sum_{j=0}^{\tau-1} \beta_j^{t,i} g_j^{t,i} - \nabla F_i(x^{t-1}) \right\|^2$$

$$= 2\mathbb{E} \left\| \frac{1}{\beta^{t,i}} \sum_{j=0}^{\tau-1} \beta_j^{t,i} (g_j^{t,i} - \nabla F_i(x^{t-1})) \right\|^2$$

$$\leq 2\mathbb{E} \sum_{j=0}^{\tau-1} \frac{\beta_j^{t,i}}{\beta^{t,i}} \left\| g_j^{t,i} - \nabla F_i(x^{t-1}) \right\|^2$$

$$\leq 2\mathbb{E} \sum_{j=0}^{\tau-1} \|g_j^{t,i} - \nabla F_i(x^{t-1})\|^2, \tag{7}$$

where we use $\|\sum_{i=1}^{n} \alpha_i x_i\|^2 \leq \sum_{i=1}^{n} \alpha_i \|x_i\|^2$ for non-negative $\sum_{i=1}^{n} \alpha_i = 1$ in the first inequality, and $\beta_j^{t,i}/\beta^{t,i} \leq 1$ in the second inequality.

For 7, we have

$$\mathbb{E}\|g_j^{t,i} - \nabla F_i(x^{t-1})\|^2 \leq 2\mathbb{E} \left\| g_j^{t,i} - \nabla F_i(x_j^{t,i}) \right\|^2 + 2\mathbb{E}\|\nabla F_i(x_j^{t,i}) - \nabla F_i(x^{t-1})\|^2$$

$$\leq 2\sigma^2 + 2L^2 \mathbb{E}\|x_j^{t,i} - x^{t-1}\|^2$$

$$= 2\sigma^2 + 2L^2 \mathbb{E} \left\| \sum_{r=1}^{j} \xi_t(g_{r-1}^{t,i}, x_{r-1}^{t,i}) \right\|^2$$

$$= 2\sigma^2 + 2L^2 j \mathbb{E} \sum_{r=1}^{j} \left\| \xi_t(g_{r-1}^{t,i}, x_{r-1}^{t,i}) \right\|^2$$

$$\leq 2\sigma^2 + 2L^2 j^2 A^2 \leq 2\sigma^2 + 2L^2 \tau^2 A^2, \tag{8}$$

where we use Assumption 2 and 1 in the second inequality. Substituting 8 into 7, we have

$$\frac{1}{M} \sum_{i=1}^{M} \mathbb{E}\|h^{t,i} - \nabla F_i(x^{t-1})\|^2 \leq \frac{2}{M} \sum_{i=1}^{M} \sum_{j=0}^{\tau-1} \left\| g_j^{t,i} - \nabla F_i(x^{t-1}) \right\|^2$$

$$\leq \frac{2}{M} \sum_{i=1}^{M} \tau \left( 2\sigma^2 + 2L^2 \tau^2 A^2 \right)$$

$$= 4\tau\sigma^2 + 4L^2 \tau^2 A^3. \tag{9}$$

Taking $D = 4\tau\sigma^2 + 4L^2\tau^2 A^3$ finishes the proof. $\qquad\square$

### B.2.2 Proof of main theorem

Now we can prove Theorem 2.

**Theorem 2.** *Under Assumptions 1, 2 and 3, if $\lambda_t$ and $\mu_t$ satisfy the condition in Theorem 1, and also if there exists $\{u_t\}$ with an exponential decay speed, such that for any $t \geq 0$, and any $g, x \in \mathbb{R}^d, \mu_t(g,x) \leq u_t$, then the global model $\{x^t\}$ satisfies:*

$$\min_{1 \leq t \leq T} \mathbb{E}\|\nabla F(x^t)\|^2 \leq \mathcal{O}\left(\frac{1}{T}\right) + \mathcal{O}\left(\underbrace{L\lambda_g^2\tau^2C^2}_{\text{weight decay error}} + \underbrace{L\lambda_g^2\tau l_*^2(\tau\sigma^2 + L^2\tau^2 A^3 + \sigma_g^2)}_{\text{client drift and data heterogeneity error}}\right),$$

*where $l_* = \max_t\{l_t\}$, with $l_t = \max_{g,x}\{\lambda_t(g,x)\}$, and $C$ satisfies $u_t\|x^t\|, u_t\|x^t\|^2 \leq C$ for any $t$.*

*Proof.* By Lemma 1,

$$x^t = x^{t-1} - \left(\lambda_g h^t + \mu_g^{t-1}x^{t-1}\right).$$

Under Assumption 1 ($L$-smooth), we have

$$F(x^t) - F(x^{t-1}) \leq -\langle \nabla F(x^{t-1}), \lambda_g h^t + \mu_g^{t-1}x^{t-1}\rangle + \frac{L}{2}\left\|\lambda_g h^t + \mu_g^{t-1}x^{t-1}\right\|^2$$

$$\leq \underbrace{-\langle \nabla F(x^{t-1}), \lambda_g h^t\rangle}_{T_1} \underbrace{-\langle \nabla F(x^{t-1}), \mu_g^{t-1}x^{t-1}\rangle}_{T_2} + \underbrace{L\lambda_g^2\|h^t\|^2}_{T_3} + \underbrace{L\mu_g^{t-1^2}\|x^{t-1}\|^2}_{T_4}.$$

We bound these four terms successively. We have

$$T_1 = -\lambda_g\left\langle \nabla F(x^{t-1}), \frac{1}{M}\sum_{i=1}^M \beta^{t,i}h^{t,i}\right\rangle$$

$$= -\lambda_g\frac{1}{M}\sum_{i=1}^M \beta^{t,i}\langle \nabla F(x^{t-1}), h^{t,i}\rangle$$

$$\leq -\lambda_g\frac{1}{2M}\sum_{i=1}^M \beta^{t,i}\left(\|\nabla F(x^{t-1})\|^2 - \|\nabla F(x^{t-1}) - h^{t,i}\|\right)$$

$$= -\frac{\lambda_g}{2M}\|\nabla F(x^{t-1})\|^2\sum_{i=1}^M \beta^{t,i} + \frac{\lambda_g}{2M}\sum_{i=1}^M \beta^{t,i}\|\nabla F(x^{t-1}) - h^{t,i}\|$$

$$\leq -\frac{\lambda_g}{2M}\|\nabla F(x^{t-1})\|^2\sum_{i=1}^M \beta^{t,i} + \frac{\lambda_g}{2M}\sum_{i=1}^M \beta^{t,i}\|\nabla F_i(x^{t-1}) - h^{t,i}\|^2$$

$$+ \frac{\lambda_g}{2M}\sum_{i=1}^M \beta^{t,i}\|\nabla F(x^{t-1}) - \nabla F_i(x^{t-1})\|^2$$

$$\leq -\frac{\lambda_g}{2M}\|\nabla F(x^{t-1})\|^2\sum_{i=1}^M \beta^{t,i} + \frac{\lambda_g\tau l_*}{2M}\sum_{i=1}^M \|\nabla F_i(x^{t-1}) - h^{t,i}\|^2 + \frac{\lambda_g}{2}\tau l_*\sigma_g^2, \quad (10)$$

where we use $2\langle a, b\rangle \geq \|a\|^2 - \|a - b\|^2$ in the first inequality and use Assumption 3 in the last inequality.

$$T_2 = -\langle \nabla F(x^{t-1}), \mu_g^{t-1} x^{t-1}\rangle \tag{11}$$

$$\leq \frac{\mu_g^{t-1}}{2}\|\nabla F(x^{t-1})\|^2 + \frac{\mu_g^{t-1}}{2}\|x^{t-1}\|^2$$

$$= \frac{\lambda_g}{2}\left(1 - \frac{1}{M}\sum_{i=1}^{M}\prod_{j=0}^{\tau-1}\left(1 - \mu_j^{t,i}\right)\right)\left(\|\nabla F(x^{t-1})\|^2 + \|x^{t-1}\|^2\right)$$

$$\leq \frac{\lambda_g}{2}\left(1 - \frac{1}{M}\sum_{i=1}^{M}(1 - u_t)^{\tau}\right)\left(\|\nabla F(x^{t-1})\|^2 + \|x^{t-1}\|^2\right)$$

$$\leq \frac{\lambda_g}{2}\left(1 - \frac{1}{M}\sum_{i=1}^{M}(1 - \tau u_t)\right)\left(\|\nabla F(x^{t-1})\|^2 + \|x^{t-1}\|^2\right)$$

$$= \frac{\lambda_g}{2}\tau u_t\left(\|\nabla F(x^{t-1})\|^2 + \|x^{t-1}\|^2\right)$$

$$\leq \frac{\lambda_g}{2}\tau u_t\|\nabla F(x^{t-1})\|^2 + \frac{\lambda_g}{2}\tau C \tag{12}$$

For the first inequality, we use $-2\langle a, b\rangle \leq \|a\|^2 + \|b\|^2$.

$$T_3 = L\lambda_g^2\|h^t\|^2 \tag{13}$$

$$\leq \frac{L\lambda_g^2}{M}\sum_{i=1}^{M}\left\|\beta^{t,i}h^{t,i}\right\|^2$$

$$\leq \frac{3L\lambda_g^2\tau l_*^2}{M}\sum_{i=1}^{M}\left\|h^{t,i} - \nabla F_i(x^{t-1})\right\|^2$$

$$+ \frac{3L\lambda_g^2\tau l_*^2}{M}\sum_{i=1}^{M}\|\nabla F_i(x^{t-1}) - \nabla F(x^{t-1})\|^2 + 3L\lambda_g^2\tau l_*^2\|\nabla F(x^{t-1})\|^2$$

$$\leq \frac{3L\lambda_g^2\tau l_*^2}{M}\sum_{i=1}^{M}\left\|h^{t,i} - \nabla F_i(x^{t-1})\right\|^2 + 3L\lambda_g^2\sigma_g^2\tau l_*^2 + 3L\lambda_g^2\tau l_*^2\|\nabla F(x^{t-1})\|^2, \tag{14}$$

where we use Assumption 3 in the last equation, and

$$T_4 \leq L\lambda_g^2\tau^2 u_{t-1}^2\|x^{t-1}\|^2 \leq L\lambda_g^2\tau^2 C^2. \tag{15}$$

Combining these together and taking expectations, we have

$$\mathbb{E}\left(F(x^t) - F(x^{t-1})\right) \leq \left(-\frac{\lambda_g}{2}\frac{1}{M}\sum_{i=1}^{M}\mathbb{E}\beta^{t,i} + 3L\lambda_g^2\tau l_*^2 + \frac{\lambda_g}{2}\tau u_t\right)\|\nabla F(x^{t-1})\|^2$$

$$+ \left(\frac{\lambda_g\tau l_*}{2} + 3L\lambda_g^2\tau l_*^2\right)\frac{1}{M}\sum_{i=1}^{M}\left\|h^{t,i} - \nabla F_i(x^{t-1})\right\|^2$$

$$+ \frac{\lambda_g}{2}\tau l_*\sigma_g^2 + \frac{\lambda_g}{2}\tau C + 3L\lambda_g^2\sigma_g^2\tau l_*^2 + L\lambda_g^2\tau^2 C^2$$

$$\leq \left(-\frac{\lambda_g}{2}\frac{1}{M}\sum_{i=1}^{M}\mathbb{E}\beta^{t,i} + 3L\lambda_g^2\tau l_*^2 + \frac{\lambda_g}{2}\tau u_t\right)\|\nabla F(x^{t-1})\|^2 + H, \tag{16}$$

where

$$H := \left(\frac{\lambda_g\tau l_*}{2} + 3L\lambda_g^2\tau l_*^2\right)D + \frac{\lambda_g}{2}\tau l_*\sigma_g^2 + \frac{\lambda_g}{2}\tau C + 3L\lambda_g^2\sigma_g^2\tau l_*^2 + L\lambda_g^2\tau^2 C^2,$$

by Lemma 2. Taking summation across all rounds $1 \le t \le T$, we have

$$\sum_{t=1}^{T} \left( \frac{\lambda_g}{2} \frac{1}{M} \sum_{i=1}^{M} \mathbb{E}\beta^{t,i} - 3L\lambda_g^2 \tau l_*^2 \right) \mathbb{E}\|\nabla F(x^{t-1})\|^2 \le F(x^0) - \mathbb{E}F(X^T) + HT. \quad (17)$$

Denote

$$\eta_t = \frac{\lambda_g}{2} \frac{1}{M} \sum_{i=1}^{M} \mathbb{E}\beta^{t,i} - 3L\lambda_g^2 \tau l_*^2. \quad (18)$$

Let $\delta > 0$ satisfy that $\eta_t > \delta$.[2] We have

$$\sum_{t=1}^{T} \delta \mathbb{E}\|\nabla F(x^{t-1})\|^2 \le \sum_{t=1}^{T} \eta_t \mathbb{E}\|\nabla F(x^{t-1})\|^2 \le F(x^0) + HT. \quad (19)$$

Therefore,

$$\min_{1 \le t \le T} \mathbb{E}\|\nabla F(x^{t-1})\|^2 \le \frac{F(x^0)}{\delta T} + \frac{H}{\delta}. \quad (20)$$

This finishes the proof. $\qquad \square$

### B.3    Validation of FedNAR

In Section B.2, we presented the proof of the main theorem. Building upon the theoretical analysis, we subsequently introduce our algorithm, FedNAR. In this section, we proceed to verify that FedNAR upholds certain properties. Additionally, we provide a specific example to illustrate the quantities utilized in the proof outlined in Section B.2.

#### B.3.1    Choices of $\lambda$ and $\mu$

Recall in the main body, we choose

$$\lambda_t(g, x) := \begin{cases} l_t \cdot A/\|g + u_t x/l_t\|, & \text{if } \|g + u_t x/l_t\| > A \\ l_t, & \text{otherwise} \end{cases}, \quad (21)$$

and

$$\mu_t(g, x) := \begin{cases} u_t \cdot A/\|g + u_t x/l_t\|, & \text{if } \|g + u_t x/l_t\| > A \\ u_t, & \text{otherwise} \end{cases}. \quad (22)$$

We now verify that they satisfy Condition 1. First if $\|g + u_t x/l_t\| > A$, we get

$$\|\lambda_t(g, x)g + \mu_t(g, x)x\| = \left\| l_t \frac{Ag}{\|g + \frac{u_t}{l_t}x\|} + u_t \frac{Ax}{\|g + \frac{u_t}{l_t}x\|} \right\|$$

$$= \frac{\|l_t Ag + u_t Ax\|}{\|g + \frac{u_t}{l_t}x\|} = l_t A \le l_* A.$$

Otherwise, if $\|g + u_t x/l_t\| \le A$, we naturally get

$$\|\lambda_t(g, x)g + \mu_t(g, x)x\| = \|l_t g + u_t x\| \le l_t A \le l_* A.$$

So our choices satisfy Condition 1.

#### B.3.2    Lower bound of $\mathbb{E}\beta^{t,i}$

As outlined in Section B.3, it is necessary to establish a lower bound for $\mathbb{E}\beta^{t,i}$. We now proceed to present a specific and explicit lower bound for $\mathbb{E}\beta^{t,i}$ in the context of FedNAR, utilizing the given choices of $\lambda$ and $\mu$.

---

[2]We will provide a specific lower bound for $\mathbb{E}\beta^{t,i}$ for FedNAR in Section B.3.

**Lemma 3.** *There exists constants $P, Q > 0$, such that for any $t \in [T]$, we have*

$$\frac{1}{M} \sum_{i=1}^{M} \mathbb{E}\beta^{t,i} \geq \frac{1}{Pt+Q} \frac{(1-u_t)(1-(1-u_t)^\tau)}{u_t} l_t,$$

*where $\beta^{t,i}$ is defined in Lemma 2, and expectation is taken with respect to random batches given $x^{t-1}$.*

*Proof.* With the definition of $\beta^{t,i}$ and $\lambda, \mu$, we have

$$\mathbb{E}\beta^{t,i} = \sum_{j=0}^{\tau-1} \mathbb{E}\beta_j^{t,i} = \sum_{j=0}^{\tau-1} \mathbb{E} \left( \lambda_j^{t,i} \prod_{r=j}^{\tau-1} \left(1 - \mu_r^{t,i}\right) \right)$$

$$\geq \sum_{j=0}^{\tau-1} \mathbb{E} \left( \lambda_j^{t,i} (1-u_t)^{\tau-j} \right)$$

$$= \sum_{j=0}^{\tau-1} \mathbb{E} \left( \min \left\{ \frac{l_t A}{\left\| g_j^{t,i} + \frac{u_t}{l_t} x_j^{t,i} \right\|}, l_t \right\} \right) (1-u_t)^{\tau-j}$$

$$= \sum_{j=0}^{\tau-1} \mathbb{E} \left( \min \left\{ \frac{A}{\left\| g_j^{t,i} + \frac{u_t}{l_t} x_j^{t,i} \right\|}, 1 \right\} \right) (1-u_t)^{\tau-j} l_t$$

$$= \sum_{j=0}^{\tau-1} \mathbb{E} \left( \frac{A}{\max \left\{ \left\| g_j^{t,i} + \frac{u_t}{l_t} x_j^{t,i} \right\|, A \right\}} \right) (1-u_t)^{\tau-j} l_t$$

$$\geq \sum_{j=0}^{\tau-1} \frac{A}{\mathbb{E} \max \left\{ \left\| g_j^{t,i} + \frac{u_t}{l_t} x_j^{t,i} \right\|, A \right\}} (1-u_t)^{\tau-j} l_t \tag{23}$$

$$\geq \sum_{j=0}^{\tau-1} \frac{A}{\mathbb{E} \left\| g_j^{t,i} + \frac{u_t}{l_t} x_j^{t,i} \right\| + A} (1-u_t)^{\tau-j} l_t. \tag{24}$$

Here 23 is from $\mathbb{E}X \cdot \mathbb{E}(1/X) \geq 1$ for positive random variable $X$, and 24 is from $\max\{a, b\} \leq a+b$. Next we give an upper bound for $\mathbb{E}\|g_j^{t,i} + \frac{u_t}{l_t} x_j^{t,i}\|$. We have

$$\mathbb{E} \left\| g_j^{t,i} + \frac{u_t}{l_t} x_j^{t,i} \right\| \leq \mathbb{E}\|g_j^{t,i}\| + \frac{u_0}{\varepsilon} \mathbb{E}\|x_j^{t,i}\|$$

$$\leq \mathbb{E}\|g_j^{t,i} - \nabla F_i(x_j^{t,i})\| + \mathbb{E}\|\nabla F_i(x_j^{t,i})\| + \frac{u_0}{\varepsilon} \mathbb{E}\|x_j^{t,i}\|$$

$$\leq \sigma^2 + \mathbb{E}\|\nabla F_i(x_j^{t,i})\| + \frac{u_0}{\varepsilon} \mathbb{E}\|x_j^{t,i}\|. \tag{25}$$

Here we denote $\varepsilon = \min\{l_t\} > 0$. For the second term in 25, by Assumption 1, we get

$$
\begin{aligned}
\mathbb{E}\|\nabla F_i(x_j^{t,i})\| &\le \mathbb{E}\|\nabla F_i(x_0^{t,i})\| + L\mathbb{E}\|x_j^{t,i} - x_0^{t,i}\| \\
&\le \mathbb{E}\|\nabla F_i(x_0^{t,i})\| + LjA \\
&\le \mathbb{E}\|\nabla F_i(x_0^{t,i}) - \nabla F(x_0^{t,i})\| + \|\nabla F(x_0^{t,i})\| + LjA \\
&\le \sum_{i=1}^{M} \mathbb{E}\|\nabla F_i(x_0^{t,i}) - \nabla F(x_0^{t,i})\| + \|\nabla F(x_0^{t,i})\| + LjA \\
&\le \sqrt{M}\sqrt{\sum_{i=1}^{M} \mathbb{E}\|\nabla F_i(x_0^{t,i}) - \nabla F(x_0^{t,i})\|^2} + \|\nabla F(x_0^{t,i})\| + LjA \\
&\le M\sigma_g + \|\nabla F(x_0^{t,i})\| + LjA \\
&\le M\sigma_g + \|\nabla F(x^0)\| + L\|x^{t-1} - x^0\| + LjA \\
&\le M\sigma_g + \|\nabla F(x^0)\| + L\lambda_g \tau A t + L\tau A,
\end{aligned}
$$

$$(26)$$
$$(27)$$
$$(28)$$
$$(29)$$

where 26 uses similar techniques as the proof of Theorem 1, 27 uses Cauchy inequality, 28 follows from Assumption 3. For the third term in 25, we get

$$
\begin{aligned}
\mathbb{E}\|x_j^{t,i}\| &\le \mathbb{E}\|x_0^{t,i}\| + \mathbb{E}\|x_j^{t,i} - x_0^{t,i}\| \\
&\le \|\nabla F(x^0)\| + L\lambda_g \tau A t
\end{aligned}
$$

$$(30)$$

Substituting 30 and 29 into 25, we get

$$
\mathbb{E}\left\| g_j^{t,i} + \frac{u_t}{l_t} x_j^{t,i} \right\| \le \underbrace{\left(1 + \frac{u_0}{\varepsilon}\right) L\lambda_g \tau A t}_{\tilde{P}} + \underbrace{M\sigma_g + \left(1 + \frac{u_0}{\varepsilon}\right)\|\nabla F(x^0)\| + L\tau A}_{\tilde{Q}}
$$

$$(31)$$

Take $\tilde{P}, \tilde{Q}$ as shown in 31, and set $P = \tilde{P}/A$, $Q = (\tilde{Q} + A)/A$. Substituting 31 into 24, we have

$$
\mathbb{E}\beta^{t,i} \ge \sum_{j=0}^{\tau-1} \frac{1}{Pt+Q}(1-u_t)^{\tau-j} l_t = \frac{1}{Pt+Q}\frac{(1-u_t)(1-(1-u_t)^\tau)}{u_t} l_t.
$$

This finishes the proof. $\square$

Additionally, we get the following natural corollary that eliminates $u_t, l_t$.

**Corollary 1.** *Denote $\varepsilon = \min\{l_t\} > 0$, then there exists a constant $P, Q > 0$, such that for any $t \in [T]$, we have*

$$
\frac{1}{M}\sum_{i=1}^{M}\mathbb{E}\beta^{t,i} \ge \frac{1}{Pt+Q}\frac{(1-u_0)(1-(1-u_0)^\tau)}{u_0}\varepsilon,
$$

*where $\beta^{t,i}$ is defined in Lemma 2, and expectation is taken with respect to random batches given $x^{t-1}$.*

*Proof.* By Lemma 3, we have

$$
\frac{1}{M}\sum_{i=1}^{M}\mathbb{E}\beta^{t,i} \ge \frac{1}{Pt+Q}\frac{(1-u_0)(1-(1-u_t)^\tau)}{u_t}\varepsilon.
$$

It remains to prove that the function

$$
f(x) := \frac{1-(1-x)^\tau}{x}
$$

decreases for $x \in (0, u_0]$. To show this, we take the derivative:

$$
f'(x) = \frac{\tau(1-x)^{\tau-1}x - 1 + (1-x)^\tau}{x^2}.
$$

Denote $g(x)$ to be the numerator of $f'(x)$, we have

$$g'(x) = -\tau(\tau - 1)(1 - x)^{\tau - 2}x + \tau(1 - x)^{\tau - 1} - \tau(1 - x)^{\tau - 1}$$
$$= -\tau(\tau - 1)(1 - x)^{\tau - 2}x \le 0.$$

Therefore,

$$f'(x) = g(x)/x^2 \le g(0)/x^2 = 0.$$

This means $f(x) \ge f(u_0)$ for any $x \in (0, u_0]$. Therefore, $f(u_t) \ge f(u_0)$ for any $t$. $\qquad\square$