# OpenReview forum: "FedNAR: Federated Optimization with Normalized Annealing Regularization"
_NeurIPS.cc/2023/Conference — NeurIPS 2023 poster_

### Official Review · Reviewer_D8ei · 2023-07-06

**Soundness:** 4 excellent
**Presentation:** 4 excellent
**Contribution:** 4 excellent
**Rating:** 7
**Confidence:** 4

**Summary:**

This paper delves into the effect of weight decay in the realm of federated optimization. The authors conduct a series of experiments highlighting how specific elements in federated learning, such as the presence of diverse data and the execution of local updates, can amplify the influence of weight decay. The paper further provides a theoretical exploration of weight decay within federated learning and introduces a novel methodology named FedNAR, which is derived from their analysis. The newly proposed FedNAR method showcases enhanced convergence speed and performance on various simulated federated tasks, and notably, it displays a heightened tolerance to complications introduced by weight decay.

**Strengths:**

1. The writing and presentation are generally of high quality, making the idea and method of this paper easy to follow.
2. The research includes sufficient empirical findings. Six different federated learning algorithms, encompassing various variants of FedAvg, are examined to investigate the impact of weight decay. Weight decay is explored across a wide range, providing clear insights into their findings.
3. The proposed algorithm, FedNAR, is well-motivated by empirical findings and appears to be a simple yet effective enhancement. It can be readily adapted to different federated learning backbone algorithms.
4. The study offers comprehensive and rigorous theoretical analysis of weight decay in federated learning. The results are novel, crucial, and hold the potential to catalyze further investigations within the federated learning community.
5. Moreover, the paper’s experimental design is comprehensive, encompassing a broad spectrum of scenarios that integrate both vision and language datasets along with a variety of hyperparameter configurations.

**Weaknesses:**

1. The bound is similar to previous works, without providing any superior bounds. This may not be a significant issue, since incorporating weight decay itself makes theoretical analysis harder and can lead to convergence challenges.
2. Though this is not my major concern, it would definitely make the paper stronger to try experiments at larger scales, e.g., GPT fine-tuning on various downstream tasks.

**Questions:**

1. As demonstrated in the paper, weight decay plays a critical role in federated learning. Is there a more effective approach to determining the optimal weight decay value for FedNAR?

**Limitations:**

As illustrated in weakness, the final bound is similar as previous works. What’s more, it would be better if the algorithms can be verified in more real-world heterogeneous data.

---

> ### Author Rebuttal · Authors · 2023-08-09
>
> ## Theoretical results
> Our theoretical analysis is constructed within the broader **non-convex** framework and is underpinned by **minimal** assumptions. This accomplishment is significant given the intricacies of federated optimization coupled with non-iid data distributions. The bounds we derive represent a **generalized** rendition of earlier findings – specifically, by setting the weight decay term to 0, we are able to recover prior outcomes. Furthermore, a sublinear bound can be achieved by setting the data heterogeneity parameter to 0, akin to previous investigations. In addition to these considerations, it's worth noting that the inclusion of weight decay introduces an additional layer of complexity, posing challenges for our theoretical analysis. This further underscores the non-trivial nature of our theoretical results.
>
> ## Additional large-scale datasets
> We've expanded our experimentation to encompass more expansive and intricate datasets, including CIFAR-100 and Tiny-ImageNet. For a comprehensive understanding of these endeavors, kindly refer to **Table 1** in the global PDF. Due to temporal and resource constraints, certain tasks such as GPT fine-tuning will be pursued in our future work.
>
> ## Hyperparameter selection
> FedNAR serves as a versatile plug-in compatible with a wide array of Federated Learning algorithms. Therefore, it suffices to employ a similar value as used in the original backbone algorithms, which can be conveniently determined through grid search, similar to practices adopted in previous studies. Consequently, there is no additional burden in the realm of hyperparameter selection for FedNAR. Notably, as evidenced by Figure 4 in the submission, FedNAR exhibits heightened robustness to the initial weight decay, implying **greater flexibility** in hyperparameter selection due to the efficacy of our co-clipping strategy.

---

> > ### Comment · Reviewer_D8ei · 2023-08-18
> >
> > Thanks for the response. It resolved most of my concerns and I will raise my score.

---

### Official Review · Reviewer_4Utv · 2023-07-07

**Soundness:** 3 good
**Presentation:** 3 good
**Contribution:** 3 good
**Rating:** 7
**Confidence:** 3

**Summary:**

The study discusses the role of weight decay in enhancing generalization performance in deep neural network optimization and in avoiding overfitting in Federated Learning (FL). The authors highlight the influence of weight decay value on FL algorithms' convergence. To mitigate this issue, Federated optimization with Normalized Annealing Regularization (FedNAR), is introduced, which modulates each update's magnitude through co-clipping of the gradient and weight decay. The algorithm shows improved accuracy in experiments on diverse vision and language datasets with various federated optimization algorithms.

**Strengths:**

- The motivation of the paper and the corresponding solution is straight-forward.
- The article is clearly articulated and readily understandable.
- This paper theoretically analyzes the impact of local training's weight decay on the convergence of federated learning.
- Experiments on both image and text datasets validate the effectiveness of the proposed method.

**Weaknesses:**

- To further validate the effectiveness of the proposed method, it would be beneficial to conduct experiments on more challenging datasets such as CIFAR100 and Tiny-ImageNet. Additionally, tests on more realistic benchmarks, like LEAF, that encompass feature disparity or imbalanced data, would provide even stronger evidence of its efficacy.

- From the results in Figure 4, it is not clear whether the proposed method is more stable for the choice of the initial value of weight decay.
In a similar vein, an ablation study on the choice of the threshold of co-clipping is required.

**Questions:**

- Is the proposed method also effective on the methods that utilize client-specific learnable parameters such as FedDC [1]?


[1] L. Gao, et al., FedDC: Federated Learning with Non-IID Data via Local Drift Decoupling and Correction, CVPR 2022.



**Limitations:**

Please refer to "Weakness" Section

---

> ### Author Rebuttal · Authors · 2023-08-09
>
> ## Additional large-scale datasets
>
> We have integrated experiments involving CIFAR-100 and Tiny-ImageNet. Kindly refer to **Table 1** in the global PDF for a detailed presentation of the outcomes. Our FedNAR consistently upholds its superior performance with about **2%~7% improvement** across all these more intricate datasets.
>
> **Experiment details:** For CIFAR-100 and Tiny-ImageNet, we ensured uniformity by retaining the same settings, parameters, and model configuration employed for the CIFAR-10 dataset, as elucidated in Section 5.1, lines 270 to 281 of our initial submission. In the case of Tiny-ImageNet, where the input size is 64x64, a minor adaptation was made to the ResNet-18 model. This modification involved transitioning the final pooling layer from avg_pool2d to adaptive_avg_pool2d to align with the altered image dimensions.
>
> **LEAF benchmarks:** In fact, the Shakespeare dataset utilized in our present submission originates from LEAF. We have employed the publicly available codes from LEAF to create the dataset, resulting in a pragmatic partitioning of the data. We will include more datasets from the LEAF benchmark in future work.
>
> ## Additional ablation studies
>
> **Stability with respect to initial weight decay:** The primary objective of Figure 4 is to illustrate the enhanced stability of FedNAR in relation to different initial weight decay selections. Upon examining Figure 4, it becomes apparent that an initial value of 0.1 is substantial, leading to a decline in performance when contrasted with the choice of 0.01 across original FedAvg, FedProx, and SCAFFOLD methodologies. However, with the incorporation of FedNAR, while there is an initial dip in accuracy compared to the 0.01 choice, accuracy gradually recuperates and even surpasses the performance of the 0.01 option. This dynamic demonstrates that FedNAR possesses the capability to autonomously rectify the detrimental impact of an excessively large initial value. In light of the observations outlined in Section 3, where marginally elevated weight decay values result in noticeable performance degradation, FedNAR emerges as a natural and efficacious remedy for this concern.
>
> **Ablations on the co-clipping threshold:** Ablation studies have been incorporated to analyze the impact of different co-clipping thresholds. The thresholds chosen for evaluation include {5, 10, 20, 40}. Please refer to **Figure 1** in the global PDF for details. The findings underscore a discernible trade-off associated with selecting the maximum norm threshold. Opting for a lower threshold results in amplified regularization, constraining each optimization step and consequently affecting performance negatively. Conversely, a higher threshold leads to less frequent occurrences of clipping, thereby yielding a milder effect. However, it's noteworthy that our results demonstrate a consistent pattern across all three backbone algorithms: regardless of the chosen maximum norm threshold, FedNAR consistently exhibits better performance.
>
> ## FedDC-NAR
> Yes, FedNAR is also **effective** on FedDC. We undertake experiments employing the publicly accessible official codes of FedDC, encompassing both CIFAR-10 and CIFAR-100 datasets. Data is partitioned across 100 clients according to a Dirichlet distribution, with an imbalance parameter from {0.3, 1, 10}. We train ResNet-18 across 500 rounds, where 20 clients are randomly selected per round. The training protocol employs a learning rate of 0.01 and an initial weight decay of 0.005. Comprehensive outcomes are presented in **Table 5** of the global PDF.

---

> > ### Comment · Reviewer_4Utv · 2023-08-21
> >
> > I thank the authors for answering my questions in detail. Most of my concerns are resolved and I will keep my original score.

---

### Official Review · Reviewer_t5c6 · 2023-07-07

**Soundness:** 4 excellent
**Presentation:** 4 excellent
**Contribution:** 3 good
**Rating:** 6
**Confidence:** 3

**Summary:**

This paper first describes an important and challenge problem in Federated Learning, that the performance of FL is very sensitive to the choice of weight decay hyper-parameter for local optimization. The authors produced data to demonstrated the sensitivity of the weight decay hyper-parameter.

This paper first outlined a general analysis framework for any client-side learning rate and weight-decay adjustment scheme, then proposed a adaptive weight decay scheme which adjust the hyper-parameter inversely proportional to the magnitude of the sum of the local gradient and the decayed-parameters. The authors provided analysis and performance guarantees for their proposed adaptive decay method. The authors conducted experiments to demonstrate the applicability of their method as a "plug-in" for different types of FL methods.

**Strengths:**

1. hyper-parameter sensitivity is often a over-looked issue in FL. Existing methods addressed this problem by a hyper-parameter optimization problem, which is costly. This methods directly make the weight decay parameter and adaptive, eliminate the need for search.

2. The authors have shown the robustness of their method as a plug-in on various FL methods, and the robustness to the initialization of the decay value, which is critical to make it actually useful.

**Weaknesses:**

In the experiments, the authors ran 5 local epochs for each algorithm. When the FL training process starts from random initialization, often single local epoch produces the best results because it limits the client drift away from the server model. I would like to see how does the proposed method perform under single local epoch.

**Questions:**

1. the authors have shown the robustness of their proposed adaptive weight decay scheme to the initial weight decay choice. But the scheme also depends on the learning rate schedule l_t, and decay schedule \mu_t, and the maximum norm A. In the experiments, the author set A = 10. How does these HPs affect the proposed method?

2. FedProx is similar to weight decay, but shrinkage to the server parameter instead of 0. Can the proposed method be used for adapting the proximal regularizer weight in FedProx?

---

> ### Author Rebuttal · Authors · 2023-08-09
>
> ## Single-epoch performance
>
> Yes, FedNAR is also **effective** in a single-epoch setting. We perform experiments using CIFAR-10 data, employing a data heterogeneity parameter of 0.3. Our approach adheres to the identical configurations, parameters, and model specifications detailed in Section 5.1, lines 270 to 281 of our original submission. The selected backbone algorithms encompass FedAvg, FedProx, and SCAFFOLD. The outcomes underscore that FedNAR consistently attains superior performance and accelerates convergence on all the aforementioned backbone algorithms. For a comprehensive overview, kindly refer to the results provided in **Table 3** of the global PDF.
>
> ## Additional ablation studies
> We expand our ablation studies to encompass learning rate scheduling, weight decay scheduling, and max norm exploration. For each of these ablation studies, the backbone algorithms include FedAvg, FedProx, and SCAFFOLD. Please refer to **Table 2** and **Figure 1** in the global PDF for the detailed results.
>
> Regarding the **learning rate**, our current submission employs exponential decay in line with prior FL research practices [24, 26]. For further ablation studies, we have additionally evaluated cosine decay and inverse linear decay, where the first is common in standard centralized training and the second is proposed in [11].
>
> Regarding **weight decay** scheduling, in conjunction with the exponential decay approach applied in our current submission, we also include cosine decay and inverse linear decay methods for an extended evaluation.
>
> Regarding the **maximum norm** parameter, we opt for values of {5, 10, 20, 40}. The outcomes demonstrate a discernible trade-off concerning the selection of this parameter. A lower maximum norm tends to amplify regularization, constricting each optimization step, which in turn adversely impacts performance. Conversely, a higher maximum norm results in less frequent clipping occurrences, yielding a diminished impact. Notably, our results indicate that across all three backbone algorithms, and for every designated maximum norm value, FedNAR consistently outperforms the baseline algorithm.
>
> ## Apply FedNAR in the proximal term of FedProx
>
> Yes, the idea of FedNAR can be seamlessly extended to the proximal term of FedProx. The implementation is similar. We proceed to replicate experiments on CIFAR-10, adhering to the same setup outlined in Section 5. To enhance the validation of this notion, we modify the coefficient of the proximal term ($\mu$) across {0.001, 0.01, 0.1}. The outcomes are presented comprehensively in **Table 4** of the global PDF.
>
> **Deeper analysis about the extension on FedProx:** The results showcase that, across all $\mu$ values, the inclusion of co-clipping in the FedProx framework yields improved performance. However, it's noteworthy that the extent of enhancement is not as conspicuous as when this technique is applied to weight decay. This disparity could be attributed to
>
> 1. The impact of FedProx loss and weight decay varies. While the FedProx loss confines the gap between the current local weight and the global weight, effectively curbing the extent of the local optimization step, this inherently mirrors the effects of the co-clipping scheme in empirical terms. Consequently, the incorporation of co-clipping doesn't provide substantial additional benefits in such a scenario.
> 2. In the context of FedProx, the proximal term represents the L2-norm of the disparity between the current local weight and the global weight. This term is anticipated to exert a relatively lesser impact compared to the current weight itself.
> 3. The co-clipping approach within FedNAR is intentionally tailored to complement weight decay, substantiated by both empirical findings in Section 3 and theoretical analyses presented in Section 4. In particular, our observations in Section 3 underline that the combination of multiple local updates and the inherent unevenness of data distribution in FL scenarios heightens the sensitivity of FL algorithms to weight decay. Additionally, the insights provided in Section 4 illustrate that co-clipping is pivotal for convergence analysis when coupled with weight decay. It's important to note that these attributes cannot be directly extrapolated to analyze the proximal term within the context of FedProx.

---

### Official Review · Reviewer_8tVP · 2023-07-27

**Soundness:** 3 good
**Presentation:** 3 good
**Contribution:** 3 good
**Rating:** 6
**Confidence:** 3

**Summary:**

The paper investigated effects of weight decay in the scenario of federated learning, especially for the stage of local updates. It was found that even subtle changes of weight decay values might lead to drastic performance drop and the observations motivated the authors to conduct convergence analysis considering the factor of weight decay. The proposed method, FedNAR, was supported by both theoretical analysis and empirical results, and demonstrated advantages when incorporated into various FL backbone algorithms.

**Strengths:**

- The paper is clearly written and well organized. In addition, the idea was motivated and inspired by an empirical study of weight decay in Section 3, which made the paper more sound.
- The authors investigated the influence of changes in weight decay of local updates, which was overlooked previously and found that the model performance was sensitive to the selection of weight decay values.
- A theoretical analysis of convergence with weight decay was presented in the paper, and supported the proposed FedNAR method

**Weaknesses:**

- It seemed that FedNAR worked well on most of federated learning backbone algorithms except for some adaptive methods such as FedAvgM and FedAdam. The accuracy was even worse after applying FedNAR to these methods. Can the authors expand on this phenomenon and provide some insights? Do adaptive methods themselves have the ability to adjust update trajectories so that adaptive weight decay might not work as expected?
- Experiments were still at a small scale and can be further improved. Currently the authors only chose one dataset CIFAR10 for image classification and one dataset Shakespeare for next-character prediction, which was not sufficiently. More large scale datasets such as ImageNet and practical ones with more realistic splits like CelebA (as suggested in the LEAF benchmark), or Apple's FLAIR dataset. Besides, details of the model architecture for each task were not clear. The selection of model structures might also affect the final performance and should be analyzed as well.

**Questions:**

- Why did FedNAR perform worse on some adaptive FL methods such as FedAdam and FedAvgM?
- Missing details about model architectures and more experiments should be included to support the proposed method.

**Limitations:**

The authors have discussed limitations in Section 6 in the paper.

---

> ### Author Rebuttal · Authors · 2023-08-09
>
> ## FedNAR and adaptive algorithms:
>
> To commence, we wish to emphasize that FedNAR is also **effective** for adaptive algorithms such as FedAdam and FedAvgm. Please consult our Table 1 and Table 2  in the original submission for reference. Among the 10 settings about FedAdam and FedAvgm covered by these tables, FedNAR demonstrates slightly inferior performance in merely 2 instances out of 10. Consequently, it remains evident that FedNAR consistently delivers enhanced outcomes within the context of adaptive methods.
>
> **Deeper analysis of adaptive algorithms and FedNAR:** These two algorithms employ update mechanisms based on momentum at the global level. This implies that the global update direction incorporates an accumulation of preceding global updates. This cumulative effect aids in rectifying current updates by counteracting deviations stemming from specific selected clients. In essence, momentum-based methodologies themselves possess attributes akin to those introduced in our FedNAR approach, which aims to control the local drift due to multiple local updates and data heterogeneity. Nevertheless, it's noteworthy that FedNAR brings forth additional enhancements, despite the inherent benefits of momentum-based techniques.
>
> ## Additional large-scale datasets
>
> Our current submission includes CIFAR-10 and Shakespeare datasets, with the Shakespeare dataset already incorporating a practical LEAF-based setup. Additionally, we have performed experiments on CIFAR-100 and Tiny-ImageNet, both of which are also widely-used benchmarks in federated learning studies. The outcomes are presented in **Table 1** of the PDF within the global response. Our FedNAR consistently maintains its superior performance, with about **2%~7% improvement** across **all** these larger and more intricate datasets. While being mindful of time and computational limitations, we intend to expand our experimentation to include CelebA and Apple's FLAIR dataset in future work.
>
> **Experiment details:** For CIFAR-100 and Tiny-ImageNet, we maintained the same settings, parameters, and model utilized with the CIFAR-10 dataset, as outlined in Section 5.1, lines 270 to 281 of our initial submission. In the case of Tiny-ImageNet, as the input size is 64x64, we made a minor adjustment to the ResNet-18 model by changing the final pooling layer from avg_pool2d to adaptive_avg_pool2d to align with the image dimensions.
>
> ## Model details
>
> For vision tasks including CIFAR-10, CIFAR-100, and Tiny-ImageNet, we use a standard ResNet-18 following previous works, as mentioned in Section 3, Line 125.
>
> For language tasks, we use a standard transformer encoder model. Specifically, we add positional encodings to the input, forward it to the 6 attention layers, and use a fully connected layer for the final prediction. Please refer to the function *class shake_transf* in the *util_models.py* in our released codes for details.

---

> > ### Comment · Reviewer_8tVP · 2023-08-18
> >
> > Thanks for the response. It resolved most of my concerns and I will keep my original score.

---

### Author Rebuttal · Authors · 2023-08-09

We express our sincere gratitude to the reviewers for their constructive and encouraging feedback. Dedicated to the continuous refinement of our work, we wish to emphasize a key contribution of our paper: **it represents the pioneering effort in systematically exploring the significance of weight decay in existing FL methods, underscored by both rigorous theoretical analysis and comprehensive experimentation - which we think are generally applicable beyond our proposed FedNAR method. Stemming from our insights, we introduced FedNAR—an essential algorithmic element that can seamlessly integrate with existing FL techniques, enabling adaptive adjustments of weight decay for enhanced convergence and model quality.** Given the time constraints of this rebuttal period, we have undertaken notable expansions in our experiments in terms of: (1) assimilating a wider and more complex range of datasets, (2) deepening our ablation studies, and (3) presenting new algorithmic innovations. The updated findings are detailed in the attached PDF.

---

### Decision · Program_Chairs · 2023-09-21

**Decision:**

Accept (poster)

**Comment:**

The paper examines the effect of weight decay in federated learning (FL) and presents some interesting findings that highlight the importance of choosing the right weight decay mechanism in various federated learning algorithms. The authors then conduct a theoretical study to derive the convergence of FL algorithms under weight decay, which serves as the motivation for a simple truncated weight decay rule. Empirical results demonstrate that the proposed weight decay rule can enhance several existing FL algorithms. However, it appears that the improvements are less pronounced for stronger algorithms (e.g., in Table 1, there is only a marginal improvement for FedAvgm, which is the strongest baseline). This raises a slight concern that the method may not lead to a significant enhancement of the state-of-the-art. Despite this minor weakness, all the reviewers believe that this paper is interesting and makes a solid contribution, so we recommend it for acceptance.